# PRMT5 as a Potential Therapeutic Target in MYC-Amplified Medulloblastoma

**DOI:** 10.3390/cancers15245855

**Published:** 2023-12-15

**Authors:** Devendra Kumar, Stuti Jain, Don W. Coulter, Shantaram S. Joshi, Nagendra K. Chaturvedi

**Affiliations:** 1Department of Pediatrics, Division of Hematology and Oncology, University of Nebraska Medical Center, Omaha, NE 69198, USA; devendra.kumar@unmc.edu (D.K.); stjain@unmc.edu (S.J.); dwcoulter@unmc.edu (D.W.C.); 2Child Health Research Institute, University of Nebraska Medical Center, Omaha, NE 69198, USA; 3Fred & Pamela Buffett Cancer Center, University of Nebraska Medical Center, Omaha, NE 69198, USA; 4Department of Genetics, Cell Biology and Anatomy, University of Nebraska Medical Center, Omaha, NE 69198, USA; ssjoshi@unmc.edu

**Keywords:** brain cancer, medulloblastoma, MYC, PRMT5 inhibitors, SDMA

## Abstract

**Simple Summary:**

Medulloblastoma is the most prevalent intracerebellar pediatric brain tumor, accounting for approximately 20% of all childhood brain tumors and over 60% of embryonal brain tumors. MYC-driven medulloblastoma has extreme metastatic potential and is often resistant to multipronged treatment. PRMT5 plays a key role in cell functions and processes in MYC-driven medulloblastoma by stabilizing the MYC protein. RMT5 inhibitors can potentially disrupt MYC’s function, impeding tumor progression and offering a target therapeutic approach to treat MYC-amplified medulloblastoma. Here, we highlight the challenges that must be addressed in future drug development.

**Abstract:**

MYC amplification or overexpression is most common in Group 3 medulloblastomas and is positively associated with poor clinical outcomes. Recently, protein arginine methyltransferase 5 (PRMT5) overexpression has been shown to be associated with tumorigenic MYC functions in cancers, particularly in brain cancers such as glioblastoma and medulloblastoma. PRMT5 regulates oncogenes, including MYC, that are often deregulated in medulloblastomas. However, the role of PRMT5-mediated post-translational modification in the stabilization of these oncoproteins remains poorly understood. The potential impact of PRMT5 inhibition on MYC makes it an attractive target in various cancers. PRMT5 inhibitors are a promising class of anti-cancer drugs demonstrating preclinical and preliminary clinical efficacies. Here, we review the publicly available preclinical and clinical studies on PRMT5 targeting using small molecule inhibitors and discuss the prospects of using them in medulloblastoma therapy.

## 1. Introduction

Medulloblastoma is the most prevalent intracerebellar pediatric brain tumor, accounting for approximately 20% of all childhood brain tumors and over 60% of embryonal brain tumors [1]. Medulloblastoma is classified into four major molecularly diverse subgroups including wingless (WNT), Sonic hedgehog (SHH, p53 mutant and p53 wild type), Group 3, and Group 4 medulloblastomas [2,3,4]. The WNT subgroup comprises approximately 10% of the medulloblastoma cases and has the most favorable clinical outcomes, with a 5-year overall survival surpassing 95% [5,6,7]. The SHH subgroup typically displays deregulation of the SHH signaling pathway and represents approximately one-third of childhood patients with medulloblastomas [2,8]. Group 3 medulloblastomas often exhibit MYC overexpression and have the most dismal clinical diagnosis of the four medulloblastoma subgroups, with a survival rate of less than 60%. MYC-driven medulloblastomas have extreme metastatic potential and are often resistant to multipronged treatment [9,10,11]. Group 4 is the most prevalent subgroup, accounting for nearly 40% of all medulloblastoma tumors, and is normally seen in children aged 5–10 years and rarely in infants [2]. Although progress has been made in understanding medulloblastoma at the molecular and genetic level, comparatively few targeted therapies have achieved clinical success. Current therapies for medulloblastoma have progressed in favor of patient survival to about 70% [8]. However, this comes with consequences, as standard treatment or medications like chemotherapy, brain and spinal cord radiation, and surgical removal leave patients at risk for permanent mental disabilities [1,12].

Post-translational modification (PTM) is one targetable regulatory mechanism of MYC and other proteins, with the potential to be developed therapeutically. While the roles of PTMs like phosphorylation [13], ubiquitinoylation [14], and acetylation [15] in controlling these proteins responsible for medulloblastoma have received significant attention, arginine methylation has only recently been investigated. Arginine methylation is one of the common PTM processes that are catalyzed by a member of the protein arginine methyltransferase (PRMT) family; this group of nine enzymes is responsible for the methylation of arginine, using S-adenosylmethionine (SAM) as a methyl group donor. The physiological control of many cellular processes, including splicing transcription and mitosis, depends on the activity of PRMT family enzymes [16]. PRMTs have also been revealed to be involved in the progression of various types of cancers [17,18]. In humans, PRMT members can be divided into various classes based on their enzymatic role, i.e., type I (PRMT1-4, PRMT6, and PRMT8) that catalyze the formation of monomethyl arginine (MMA) and asymmetric dimethyl arginine (ADMA); type II (PRMT5 and PRMT9) that catalyze the formation of MMA and symmetric dimethyl arginine (SDMA); and type III (PRMT7) which is responsible for the formation of MMA [19]. As the most prevalent type II SDMA methyltransferase, PRMT5 forms a heterotetrametric complex with a protein called methylosome protein 50 (MEP50) that can catalyze symmetric demethylation of various histone and non-histone proteins [20]. Remarkably, PRMT5 was proven to regulate the function of glioma-associated oncogene homolog 1(GLI1) protein in an SHH-responsive cell line [21]. PRMT5 also represents a requisite driver of tumor progression in SHH-medulloblastoma and MYC-amplified medulloblastoma [22,23]. During conversion to malignancy or metastasis, PRMT5 acts as an oncogene. PRMT5 enzyme inhibition or its catalytic depletion frequently reduces or halts cellular proliferation, while its hyperexpression leads to hyper-proliferation [24,25,26]. Consequently, PRMT5 is emerging as a novel target for the treatment of various cancers, including medulloblastoma.

Recently, PRMT5 inhibitors have been credited with inhibiting the growth of cancerous cells in vitro and in vivo. Various PRMT5 inhibitors with different functions have undergone clinical trials for the treatment of advanced cancer or recurrent solid tumors [27,28]. The effects of PRMT5 inhibition on cancerous cells’ proliferation, invasion, and migration can contribute to anti-cancer efficacy [29,30]. This review explores the current knowledge of the effectiveness of PRMT5 inhibitors in preclinical and preliminary clinical settings, which may aid in understanding how to treat MYC-amplified medulloblastoma more effectively and safely.

## 2. PRMT5 Structure, Function, and Localization

### 2.1. Structure

PRMT5 is a primary type II arginine methyl transferase that forms a prominent methylosome complex with distinctive binding oligopeptides, such as the WD (Trp-Asp) repeat-containing 50-kilodalton methylosome protein (MEP50). PRMT5 requires the existence of diverse substrate adapters such as rio-domain-containing protein 1 (RioK1), chloride channel nucleotide-sensitive 1A protein (pIC1n) and cooperator of PRTM5 (COPR5) to detect and catalyze the SDMA on histone and non-histone proteins via PTMs [31,32,33]. PRMT5′s structure consists of a triphosphate isomerase (TIM) barrel, an intermediate Rossmann-fold, and a C-terminal β-barrel [34]. Four PRMT5 units generate a hetero octameric complex by binding with four MEP50s (Figure 1). Studies have demonstrated that PRMT5 alone has minimal methyltransferase activity; it must be complexed with MEP50 to achieve normal catalysis of SDMA on proteins [35]. This could be because MEP50 enhances the stability of PRMT5 for a long time by binding with proteins and acting as a metastable cofactor.

### 2.2. Function

PRMT5 plays a key role in cell functions and processes by regulating the methylation of cellular proteins, which affects oncogenic cellular processes such as cell proliferation and differentiation [29,30,36]. PRMT5 regulates these processes by modifying gene expression to stabilize histones H4R3, H3R2, H3R8, and H2AR3 and non-histone proteins via the SDMA process [37,38]. An extensive range of nonhistone proteins have also been revealed as PRMT5 substrates, including androgen receptor (AR), EGFR, GATA4, C-MYC, N-MYC, IL-2, E2F1, GM130, HOXA9, KLF4, KLF5, NOTCH, NFkB(p65), PDCD4, POLR2A, P53, RAF proteins, SPT5, SREBP1a, Sm proteins, nucleolin, and others [12,36,39,40,41,42,43,44,45,46,47,48,49,50,51,52]. In addition, some substrates such as certain nonhistone oncogenic transcription factors are symmetrically dimethylated by PRMT5. PRMT5-regulated cellular processes are shown in Figure 2. The importance of arginine methylation by PRMT5 in cancer progression has only recently become apparent [17]. PRMT5 knockout mice exhibited embryonic lethality, demonstrating the role of PRTM5 in embryonic development and crucial biological functions. In mouse embryonic stem cells (ESCs), PRMT5 maintains pluripotency, whereas in human ESCs, it influences only proliferation [53,54]. PRMT5 is needed for neural stem cell persistence and its deletion causes premature death of the mouse by disrupting the development of the central nervous system [55]. PRMT5 promotes SWI/SNF-mediated chromatin remodeling and controls the process of myogenesis. Deletion of PRMT5 causes an obstacle in developmental processes, uncontrolled proliferation, and impairment of adult tissue differentiation [53,54,55,56,57,58]. Notably, PRMT5 is overexpressed in a number of cancers, including melanoma, multiple myeloma, lymphoma, glioblastoma, breast, lung, pancreas, prostate, ovarian, and colorectal cancers, and high expression of PRMT5 often correlates with poor patient clinical outcomes [38,59]. Organ-specific functions of PRMT5 are shown in Table 1. The higher expression of PRMT5 in cancer is thought to epigenetically suppress tumor suppressor and cell cycle genes [17,60]. Recently, the association of PRMT5 with MYC was found in numerous cancers, including brain tumors such as glioblastoma; this association creates abnormalities in MYC function [61,62,63]. Consequently, PRMT5 has been recognized as an oncogenic function and has received extensive interest as a potential target for better clinical outcomes. To this end, numerous potent therapeutic agents have been developed to inhibit PRMT5 and their antitumor effects are now being assessed in preclinical models and clinical trials [27,64].

### 2.3. Localization

Cytosolic and nuclear localization of PRMT5 helps to determine its role in the cell. PRMT5 is predominantly localized in the cytoplasm in lung [65], prostate [66], and melanoma cancer [67]. Diffused cellular localization of PRMT5 was confirmed in both the cytoplasm and the nucleus of brain tumor glioblastoma cells [61]. Cytoplasmic and nuclear localization of PRMT5 has also been confirmed in various preclinical mouse models and primary human cancer tissues [68]. In adult mice, PRMT5 is expressed predominantly in the nucleus of the neurons in the cerebrum and spinal cord [55]. Han et al. demonstrated the high expression of PRMT5 as a marker of malignant progression in glioblastoma and its crucial role in tumor growth [63]. Our lab recently analyzed the localization of the PRMT5 in tumor tissues of medulloblastoma patients as well as in MYC-amplified cell lines. PRMT5 demonstrated predominantly nuclear localization in both HD-MB03 and primary tumor cells [22].
cancers-15-05855-t001_Table 1Table 1Organ-specific roles of PRMT5.OrganCellular FunctionMechanismReferencesBrainCell cycle progression, apoptosisAltered expression and stability of MYC[22]
Phase separation Methylation of FUS [69]
GSK3β-NF-kβ signalingAltered expression of E2F1[70]
HTT toxicityAltered expression of HTT[71]
AKT-ERK signaling, cell cycle progression Altered expression of PTEN[72]
Cell cycle progression, stemnessAltered RNA Splicing[73,74]
mTOR signaling Methylation (hnRNPA1)[75]
DNA instability responseAltered expression of RNF168[76]
Cell migration, cell cycle progression, and apoptosisAltered expression of LRP12[62]
AKT signaling and metastasisMethylation of PKB[77]LungsMetastasisAltered expression of EMT genes [78]
MetastasisAltered expression of SHARPIN[79]
MetastasisAltered expression of FGFR3/miR-99 family[80]
MetastasisMethylation of KLF5[81]LiverLipid metabolismMethylation of SREBP [47]
ERK signalingAltered expression of BTG2[82]
PRMY5 deprivationPRMT5 activity of LINC01138[83]
WNT-β-Catenin signalingAltered cofactor binding of LYRIC[84]SpleenNAAltered stability of MYC[85]PancreasGlucose metabolism, cell cycle progressionAltered stability of MYC[86,87]Bone Type I interferon signaling Altered expression (interferon gene)[88,89]ProstateAR, ERG signaling Altered methylation (AR)[90,91]OvaryNAAltered methylation (E2F1)[92]HeartTranscriptional activityMethylation (GATA4)[45]Breast Stemness Altered expression of C-MYC, KLF4, and OCT4[93]
StemnessAltered expression of FOXP1[94]
Metastasis and invasionAltered expression of AKT genes[78]
Metastasis and AKT signalingMethylation of AKT[95]
Cell cycle progressionMethylation of KLF4[96]
Cell migrationMethylation of ZNF326[97]
NAMethylation of PDCD4[98]Abbreviations: AKT-ERK, alpha serine/threonine-protein-extracellular-regulated kinase; AR, androgen receptor; E2F1, E2 promoter binding factor 1; EMT, epithelial–mesenchymal transition; ERG, ETS-related gene; FGFR3, fibroblast growth factor 3; FOXP1, forkhead box protein P1; FUS, fused in sarcoma; GSK3β-NF-kβ, glycogen synthase kinase; hnRNPA1, heterogeneous nuclear ribonucleoprotein A1; HTT, huntingtin protein; KLF4, Kruppel-like factor 4; LRP12, low-density lipoprotein receptor-related protein 12; LINC01138, long non-coding RNA; miR-99, microRNA99; mTOR, mammalian target of rapamycin; OCT4, octamer binding protein 4; PKB, protein kinase B; PDCD4, program cell death protein 4; SREBP, sterol regulatory element-binding protein; ZNF326, zinc finger protein 326.


## 3. PRMT5 Association with MYC-Driven Medulloblastoma

Epigenetic deregulation plays a key role in medulloblastoma tumorigenesis, especially in aggressive Group 3 and Group 4 medulloblastomas [99,100,101,102], where germline mutations in known cancer predisposition genes are rare. Indeed, epigenetic deregulator or chromatin modifiers, including histone acetylase or methylation/methyltransferase activities, are very common in Group 3 and 4 medulloblastomas compared to other subgroups. This emphasizes the need to discover and understand the pertinent mechanisms of epigenetic regulation or PTMs and the corresponding therapeutic targets. We recently reported that PRTM5 is a critical regulator MYC oncoprotein in an MYC-amplified (Group 3) medulloblastoma [22]. We found that high levels of PRMT5 not only mirror MYC expression but also correlate with poor outcomes in Group 3 medulloblastoma patients. Mechanistically, we showed that PRMT5 stabilizes the MYC protein by physically interacting with it, raising the intriguing possibility that PRMT5 can regulate MYC function at both the transcriptional and translational/post-translational levels. The exact MYC oncogenic programs regulated by PRMT5 in medulloblastoma are largely unknown. Therefore, exploring the regulation of MYC-driven oncogenic progresses by PRMT5 is crucial to identify effective therapeutics for these high-risk patients.

The involvement of PRMT5 has been verified in the epigenetic regulation of chromatin complexes following interaction with numerous proteins, including transcription factors [42], and their activities are dysregulated in various cancers [59]. In recent studies, high levels of PRMT5 and MYC corelate with glioma malignancy [61,62,63]. Furthermore, PRMT5 is physically associated with N-MYC (an MYC homologue) and enhances the stability of N-MYC in neuroblastoma cells [51]. Nonetheless, the function of PRMT5 and its interaction with MYC in MYC-driven medulloblastoma have not been fully investigated. Favia et al. reported that the association of PRMT1 and PRMT5 with MYC in glioblastoma stem cells resulted in MYC being dimethylated symmetrically and asymmetrically by both enzymes, respectively [103]. MYC-driven cellular processes resulting from symmetric dimethylation by PRMT5 are shown in Figure 3. The colocalization of PRMT5 and MYC suggests that PRMT5 forms a complex with MYC and supports its stabilization in MYC-amplified medulloblastoma cells. This physical interaction of PRMT5 and MYC implies a potential role of PRMT5 in medulloblastoma tumorigenesis.

Highly expressed PRMT5 stabilizes MYC and promotes its expression in medulloblastomas. Studies support the predictive value of PRMT5 overexpression as a biomarker for aggressive tumorigenesis in cancer patients. Knockdown of PRMT5 in medulloblastoma cells suppresses cell growth by diminishing MYC stability, supporting the functional role of the PRMT5–MYC interaction complex in medulloblastoma [22]. Since MYC and PRMT5 co-expression and colocalization were observed in the nucleus, PRMT5 could also regulate MYC function at the transcriptional level. Further studies are needed to investigate PRMT5’s roles in the regulation of the transcription and translation of MYC.

PRMT5 is a stemness factor crucial in maintaining the balance between quiescence, proliferation, and generation for cancer stem cells and non-cancer cells. The role of PRMT5 in stemness has been demonstrated in embryonic (ESCs) and neural stem cells (NSCs) [53,72,104]. Provided that NCCs or cancer stem cells have a great influence on medulloblastoma recurrence and tumorigenesis, there might be a role for PRMT5 in regulating the self-renewal of tumor initiation in medulloblastoma. Recently, the methylation of stemness factor KLF-4 (Kruppel-like factor-4) by PRTM5 was shown in breast cancer [96]. This methylation leads to KLF4 protein stabilization, promoting tumorigenesis. In another study, researchers synthesized a novel compound that has the potency to inhibit PRMT5, disrupt the interaction of PRMT5 and KLF4, and suppress breast cancer development [105]. KLFs are evolutionarily conserved zinc-finger-associated transcription factors with distinct regulatory functions in cell growth, proliferation, and differentiation. Moreover, PRMT5 interacts with KLF5 (another member of KLF family proteins) and accelerates its dimethylation, a mechanism that depends on methyltransferase activity [81]. Further investigation to understand the mechanism of PRTM5–KLF4/KL5 interactions could uncover another new strategy to elucidate therapeutic targets for MYC-amplified medulloblastoma.

## 4. Potential Inhibitors of PRMT5

PRMT5 inhibitors have been proven to prevent the growth of cancerous cells in vitro and in vivo. Many PRMT5 inhibitors have entered clinical trials for the treatment of multiple types of cancer [34,106,107]. The pharmacological effects of these inhibitors with their targets in various cancers are summarized in Table 2, and details about corresponding clinical trials are given in Table 3.

### 4.1. JNJ-64619178

JNJ-64619178 (International Patent Number: WO/2017/032840 A1) is a potent PRMT5 inhibitor that irreversibly binds to the SAM pocket of the PRMT5/MEP50 and establishes a short kinetic constant of target unbinding, resulting in prolonged trapping of PRMT5/MEP50 in an inactive transition that impedes arginine methylation of histone proteins to reduce cellular proliferation [130,131]. The pharmacokinetic (PK) profile of JNJ-64619178 on a single post-oral dose (PO; 10 mg/kg) and intravenous (IV; 2.5 mg/kg) administration led to a low clearance (CL = 6.6 mL/min/kg) in mice and reasonable oral bioavailability (F = 36%). It is presently in clinical trials (NCT03573310) for patients with advanced solid tumors, non-Hodgkin lymphoma, and lower-risk myelodysplastic syndrome [28,132]. Initial clinical results revealed evidence that JNJ-64619178 has manageable toxicity and antitumor activity at a dose of 1.5 mg QD [132]. A phase 1 dose escalation study involving 90 patients was conducted to identify recommended phase 2 dose (RP2D) levels for JNJ-64619178. Based on safety, clinical activity, and PK and pharmacodynamic (PD) outcomes, two RP2Ds (1.5 mg intermittently and 1 mg once daily) were selected to inhibit PRMT5 activity in patients with cancerous tumors [133].

### 4.2. PF06939999

PF06939999 is another potent, selective SAM-competitive inhibitor whose complete mechanism is still unknown. PF06939999 displayed superior in vitro and in vivo antitumor activity with concomitant loss of SDMA [109]. The drug sensitivity to PF06939999 in non-small cell lung cancer (NSCLC) is associated with signaling pathways involving MYC, cell cycles, and spliceosomes and with mutations in splicing factors. The PK profile of PF06939999 in a single dose (PO, 10 mg/kg; IV, 2 mg/kg) revealed a reasonable plasma clearance (CL = 40 mL/min/kg) and steady-state volume of distribution (Vss 3.8 L/kg) in rodents with moderate oral bioavailability (F = 40%). A phase I dose escalation clinical trial (NCT03854227) showed promising results in patients with various cancers, including NSCLC, head and neck squamous cell carcinoma (HNSCC), and others [132,134]. The results of the NCT03854227 were described in the ASCO annual meeting in 2021 [135].

### 4.3. EPZ015666

EPZ015666 is a selective substrate-competitive inhibitor of PRMT5 with potential antiproliferative and antineoplastic activity [110,136]. Previously, it was known as GSK53235025 [106]. This inhibitor was first employed in mantle cell lymphoma. It was also used in multiple myeloma and medulloblastoma [137]. The efficacy of EPZ015666 was determined on the three MYC-amplified medulloblastoma cell lines (HD-MB03, D-283, and D-341). Medulloblastoma cells were treated with EPZ015666 in a dose-dependent manner for 72 hr and the results of cell growth assays confirmed that EPZ015666 induced growth inhibition directly proportionate to the dose in all MYC-driven medulloblastoma cell lines at a low micromolar potency, with an IC50 of 1.5–2.5 μM [22]. The PK profile of EPZ015666 in a single dose (oral, 100 mg/kg) revealed a low plasma clearance and a satisfactory brain distribution in mice. EPZ015666 significantly downregulates the higher expression of PRMT5 and MYC in medulloblastoma cells [22], suggesting it has therapeutic potential for MYC-driven medulloblastoma.

### 4.4. GSK3326595

GSK3326595 is a selective substrate-competitive PRMT5 inhibitor with potential antitumor and antiproliferative activity, which has shown efficacy in various tumor models [77]. Two clinical trials [138,139,140] are assessing this compound in patients with solid tumor cancers, primarily adenoid cystic carcinoma and colorectal and breast cancer. A phase I (NCT02783300) clinical trial is underway to assess the safety, PK, and PD in adults with solid tumors. General adverse events in this study were common but mild. Another clinical trial (NCT03614728) on reverted or refractory myelodysplastic syndrome (MDS), chronic myelomonocytic leukemia (CMML), and acute myeloid leukemia (AML) from MDS is active [141]. A third trial, designed to evaluate the drug in patients with early breast cancer, has been completed but no results have been posted yet [142].

### 4.5. AMG 193

AMG 193 is a methylthioadenosine (MTA)-cooperative PRMT5 inhibitor that specifically targets the MTS-bound state of PRMT5 [119]. This state is enhanced in methylthioadenosine phosphorylase (MTAP)-null tumors. AMG 193 has shown potential inhibition in a patient-derived xenograft model as well as MTAP-null cancer cell lines. NCT05094336, a first-in-human (FIH), open-label, multicenter phase I/II trial, is enrolling patients with MTAP-null NSCLC to evaluate the safety, tolerability, PK, PD, and efficacy of AMG 193. Docetaxel is used as a combination drug for this clinical trial [143]. However, MTAP-null mutations are still not identified in medulloblastoma, so AMG 193 might be less relevant for treatment.

### 4.6. PRT543

PRT543 is a potent PRMT5 inhibitor that inhibits the methyltransferase activity of PRMT5 by selectively binding to it, causing potent inhibition of cellular proliferation and SDMA formation in various cancerous cell lines [107,144]. PRT543 is currently under assessment in a phase I (NCT03886831) clinical trial that has been completed in patients with advanced solid tumors and hematologic malignancies. The purpose of the study was to define a safe dose and timetable for consumption in successive developments of PRT543. This dose-escalation, open-label study initially provided favorable results. Target engagement was confirmed by measuring serum SDMA. Phase I dose escalation and expansion studies are continuing to enroll patients.

### 4.7. PRT811

PRT811 is a selective and orally bioavailable PRMT5 inhibitor that passes the blood–brain barrier and shows effectiveness in high-grade glioma. PRT811 is currently under evaluation in a multicenter, open-level, phase I clinical trial (NCT04089449) in patients with central nervous system (CNS) lymphomas, recurrent high-grade gliomas, and advanced solid tumors. PRT811 has excellent PK properties in multiple preclinical species with a >two-fold higher brain vs. plasma exposure in rodents. PRT811 quickly penetrates the blood–brain barrier in rodents with higher exposure. PRT811 inhibits SDMA and cell proliferation of brain tumor cells [145]. PRT811 is broadly active against brain cancer cells and cancers with high rates of brain metastases [145]. Initial data were presented at the AACR-NCI-EORTC conference held in 2021 [121].

### 4.8. TNG908

TNG908 is also an MTA-cooperative PRMT5 inhibitor. The MTA-cooperative binding process has demonstrated the synthetic lethal relationship between MTAP losses and PRMT5 inhibition. TNG908 demonstrated a 15-fold higher potency in MTAP-null cancer cell lines. Pharmacokinetically, it is not a substrate of efflux transporters like Pgp and BCRP, which is a favorable predictor of the ability to cross the blood–brain barrier [122]. However, we still need to measure its exposure in the brain and determine the Kp. The PD properties of NTG908 allow for PRMT5 inhibition, which was confirmed by decreased levels of SDMA-modified proteins in a dose-dependent manner in a glioblastoma xenograft model. TNG908 demonstrated antineoplastic activity against MTAP-null selective tumors in various xenograft models, including tumor regression in a model representing NSCLC and cholangial and urothelial carcinomas [14,146]. One clinical trial (NCT05275478) is going to recruit patients with locally advanced solid tumors.

### 4.9. MRTX1719

MRTX1719 is another MTA-cooperative inhibitor of PRMT5. MRTX1719 catalytically binds the PRMT5–MTA complex and stabilizes it in an inactive form. In vitro, MRTX1719 demonstrates a long-lasting therapeutic effect in MTAP cells. Additionally, in vivo studies verified that MRTX1719 demonstrates potent and enduring inhibition of PRMT5 in a MTAP-deleted tumor xenograft model, reducing its SDMA activity [123]. An ongoing phase I/II clinical trial (NCT05245500) is evaluating the safety, tolerability, PK/PD, and antineoplastic activity against advanced and metastatic solid tumors. Preliminary data have been presented, including objective responses in patients with mesothelioma, MTAP-deleted melanoma, gallbladder adenocarcinoma, NSCLC, malignant peripheral nerve sheath tumors, solid tumors, and pancreatic adenocarcinoma [124].

### 4.10. LLY 283

LLY 283 has the potential to inhibit PRMT5 by binding competitively to the SAM binding site of PRMT5 as a cofactor competitive inhibitor [125]. LLY 283 can efficiently permeate the blood–brain barrier, as the compound is eliminated more quickly from the plasma than from the brain [74]. LLY 283 significantly decreases SDMA levels in cancerous cells. Pharmacologically, LLY 283 suppressed the growth of glioblastoma cell cultures derived from a cohort of 46 patients. Importantly, LLY 283 has shown significant survival benefits in mice implanted with a patient-derived xenograft (PDX), a preclinical orthotopic model of glioblastoma, even though more preclinical and clinical studies are warranted. LLY 283 showed a satisfactory PK profile, including a high metabolic stability and moderate permeability with oral bioavailability (F = 50%), which makes it an effective probe molecule for in vivo assessment.

### 4.11. Compound1a

Compound1a has also been recognized as a potential PRMT5 inhibitor. It binds allosterically to PRMT5 and competes with SAM at the binding site. Compound1a has previously been described as a human β-secretase (BACE1) and BACE2 inhibitor [147]. It demonstrated targeted effectiveness and cell-based inhibition of MCF7 cells, based on quantitation of symmetrically dimethylated nuclear protein levels. Compound1a makes an enzyme–substrate complex to bind with the co-crystal system of PRMT5–MEPP50. This complex reveals that a distinctive binding mode and considerable structural changes in the backbone of PRMT5 result from SAM-competitive inhibition [106,127,137].

### 4.12. CMP5

CMP5 is another molecule identified as a PRMT5 inhibitor [72]. By reducing the recruitment of PRMT5 in the glioblastoma cell line, it reduces the methylation of histone. However, it does not demethylate histones that have already been methylated by PRMT5. CMP5 has shown the capability to control differentiated and undifferentiated cancerous cell populations [148] and induce senescence and apoptosis of cancerous cells [136]. CMP5 has been shown to have anti-cancer efficacy against a glioblastoma xenograft model. In preclinical PK studies, CMP5 was revealed to accumulate in brain tissue without causing toxicity [149]. Chromatin histone methylation in the promoter region of DKK1 and DKK3 was hindered by CMP5-based inhibition of PRMT5, which decreased the expression of cyclin D1 and SUBRVIVIN [137]. Overexpression of cyclin D1 is directly linked to cancer progression.

### 4.13. GSK591

Previously known as EPZ015866, GSK591 was characterized as a potent inhibitor of PRMT5, including in vivo [27,110]. Proliferation of CRC cells is directly related to PRMT5 activity, and the inhibition of PRMT5 activity by GSK591 can stop proliferation and cell cycle progression. GSK591 significantly decreases SDMA in a dose-dependent manner and decreases the viability of neuroblastoma cell lines in a nanomolar range [77]. In one study, GSK591, in combination with LLY283, showed substantial survival benefits in an orthotopic PDX mouse model, although more preclinical and clinical studies are warranted in the future [74].

### 4.14. PRT382

PRT382 is a selective PRTM5 inhibitor with an adenosine backbone structurally similar to other PRMT5 inhibitors (JNJ-64619178, PF-06855800, LLY-283) [85]. PRT382 appears to have a similar SAM-competitive mechanism and optimal enzymatic kinetics in vitro that produces an IC50 of 2.8 nM with PRMT5/MEP50. It reduces SDMA with an IC50 of 27 nM and has antiproliferative activity in leukemia and lymphoma cancerous cell lines [85]. PRT382 displays low clearance and a high oral bioavailability in preclinical models. It is important to delineate the distribution of PRT382 in the brain and its efficiency in crossing the blood–brain barrier.

### 4.15. JBI-778

JBI-778 is a potent and strong inhibitor of PRMT5 [128] that reduces SDMA at an effective concentration of <10 nM. It exerts a strong antiproliferative activity in selected cell lines like NSCLC, neuroblastoma, glioblastoma, and medulloblastoma, with an IC50 ranging from 27 to 700 nM. JBI-778 can penetrate the blood–brain barrier with very high brain exposure in rodents, and it showed a favorable oral bioavailability in mice (F = 66%), rats (F = 52%), and dogs (F = 47%). JBI-778 showed strong tumor growth inhibition in a glioblastoma orthotopic model that mimics human GBM, with a significant extension in survival. Its differentiated mechanism makes it a potential option to treat brain metastasis cancers. Jubilant Therapeutics has received FDA clearance for an investigational new drug application (IND) to recruit patients for a phase I/II clinical trial for the assessment of safety, optimal doses, and PK properties of JBI-778 in patients.

### 4.16. SH3765

SH3765 is an orally bioavailable selective inhibitor of PRMT5 with antineoplastic activity that binds to PRMT5 and inhibits its methyltransferase activity at both monomethylated and dimethylated arginine residues in histone proteins. SH3765 modulates the gene expression implied in several cellular processes and decreases the growth of rapidly proliferating cells, including cancer cells. A phase I clinical trial (NCT05015309) will begin shortly to assess the safety, tolerability, and PK profile in patients with solid tumors with advanced malignancy to finalize the maximum tolerated dose (MTD) and RP2D [129].

### 4.17. SCR6920

SCR6920 is another orally bioavailable selective PRMT5 inhibitor with antiproliferative activity. A phase I open-label multicenter clinical trial (NCT05528055) will assess the dose escalation, safety, tolerability, and preliminary efficacy of SCR6920 in patients with advanced malignant tumors following oral administration. The dose-limiting toxicity must be the priority of this trial. The purpose of this clinical trial is to find the MTD, identify the RP2D, and accrue preliminary efficacy data in the participants [129].

## 5. Future Perspective and Conclusions

PRMT5-regulated oncogenes, such as C-MYC and N-MYC, are often deregulated in medulloblastomas. PRMT5 symmetrically dimethylates many proteins to regulate their stability and control activity in subcellular locations. The inactivation of PRMT5 has been shown to prevent MYC-driven lymphomagenesis [26]. PRMT5 is highly overexpressed in multiple aggressive metastatic cancers [150]. The promising role of PRMT5 in solid tumors has provoked the discovery and development of candidate drugs targeted to PRMT5 that display competitive and uncompetitive inhibition of SAM-mediated enzymatic activity. Critically, it is known that the genomic instability and catalytic activity of PRMT5 in MYC-amplified medulloblastoma cells decrease cell proliferation and induce apoptosis, which supports PRMT5 inhibition as a therapeutic option for MYC-driven medulloblastoma. More studies are needed to understand the mechanisms of PRMT5 overexpression that may cooperate with recurrent genomic lesions to contribute to medulloblastoma progression. Exploring the mechanisms of interaction between PRMT5 and MYC should give us further insights into how the two are engaged in promoting medulloblastoma aggressiveness. In addition, our in vitro and in vivo analyses of the inhibition of PRMT5, either with gene therapy or pharmacologically active small molecules as PRMT5 inhibitors, have indicated the potential of the PRMT5–MYC axis as a novel therapeutic approach in MYC-amplified medulloblastoma. As mentioned, the function of PRMT5 contributes to various physiological cellular processes to preserve cancer phenotypes and promote cancer progression in various cancer types. There is a strong rationale that the perturbation of PRMT5 can be a broadly effective means to treat cancer.

The development of PRMT5 inhibitors to achieve supportive efficacy is still in progress, and many PRMT5 inhibitors developed as SAM-competitive drugs are under clinical evaluation. However, most PRMT5 inhibitors have unwanted cytotoxicity in non-cancerous cells and in healthy tissues in clinical settings. This issue should be addressed in the context of MYC-amplified medulloblastoma. PRMT5 inhibitors could affect the signal transduction and reinstate the function of tumor suppressors via inhibition of the SDMA process. Inhibitors targeting PRMT5-mediated dimethylation may be attractive as single agents or in combination with other agents targeting MYC-amplified medulloblastoma to induce a durable response and prevent or delay acquired resistance. As PRMT5 is essential for normal cellular processes, clinical evaluation of the PRMT5 inhibitors in cancer therapy must carefully examine safety outcomes [151].

Limitations: Most investigational drugs, including some PRMT5 inhibitors, are prevented from efficiently entering into the brain. PRMT5 inhibitors are apparently transported back to the systemic circulation by the multidrug efflux pump action of proteins like P-glycoprotein (P-gp) [152,153]. Insufficient transport of drugs into the brain leads to a diminished therapeutic effect and aggravated organ toxicity side effects due to the deposition of the drug in other organs and tissues. Hence, novel PRMT5 inhibitors with satisfactory PK and PD profiles deserve additional refinement to confer more potent PRMT5 inhibition so they can be administered in minimum doses with the maximum effective concentration (MEC). It is urgent to address the issue of brain-targeted therapeutics by developing effective and safe drug delivery strategies for PRMT5 inhibitors. Several PRMT5 inhibitors are under clinical evaluation and are currently being examined in cancer patients with solid tumors, including neuroblastoma and glioblastoma. First-generation PRMT5 inhibitors cause side effects, including anemia, neutropenia, and thrombocytopenia [107,133], which can limit the capacity to reach the dose and exposure necessary to drive tumor regression in patients. The outstanding question is which PRMT5 targets should be traced in MYC-driven medulloblastoma to monitor and predict the response. GSK3326595 showed a significant inhibitory effect on the growth of MYC-driven medulloblastoma cell lines at a low micromolar potency and showed PRMT5 downregulation. Thus, GSK3326595 could potentially be further investigated at the clinical level for MYC-driven medulloblastoma. Interestingly, TNG908 has been advanced in a clinical trial as a PRMT5 inhibitor that is able to penetrate the blood–brain barrier. Two other compounds, LLY283 and CMP5, have shown favorable PK properties, along with brain distributions that suggest efficient penetration of the blood–brain barrier. JBI-778 demonstrated strong tumor growth inhibition in a glioblastoma orthotopic model and a favorable oral bioavailability. Some PRMT5 inhibitors have a very low IC50 in vitro but cannot cross the blood–brain barrier. MTA-cooperative PRMT5 inhibitors have favorable PK properties and efficiently penetrate the blood–brain barrier. However, MTAP-null mutations have still not been detected in medulloblastomas, so MTA-cooperative PRMT5 inhibitors might have less relevance compared to other PRMT5 inhibitors. JNJ-64619178 and PF06939999 have 30–40% oral bioavailability, although they are very effective in vitro. Enhancing blood–brain barrier penetration is crucial to improving the therapeutic efficacy and lowering toxicity.

Combining PRMT5 inhibitors with other drugs (e.g., chemotherapy, immune checkpoint inhibitors, or anti-EGFR drugs) in medulloblastoma treatment may hold promise by synergistically targeting cancer cells through different mechanisms. This approach may enhance the treatment efficacy, overcome drug resistance, and reduce the potential side effects associated with higher doses of individual agents. Research suggests that combining PRMT5 inhibitors with standard chemotherapy regimens could provide a more comprehensive and effective strategy for specific cancer types, including those with MYC amplification [154,155,156,157]. Clinical trials are underway to further explore the safety and efficacy of such combination treatments.

This review provides a comprehensive survey of possible PRMT5 inhibitor therapeutics to treat MYC-amplified medulloblastoma and highlights the challenges that must be addressed in future drug development.

## Figures and Tables

**Figure 1 cancers-15-05855-f001:**
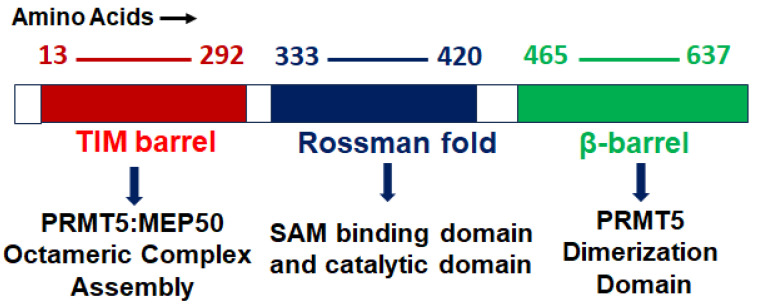
PRMT5 protein structure: structural and functional domains.

**Figure 2 cancers-15-05855-f002:**
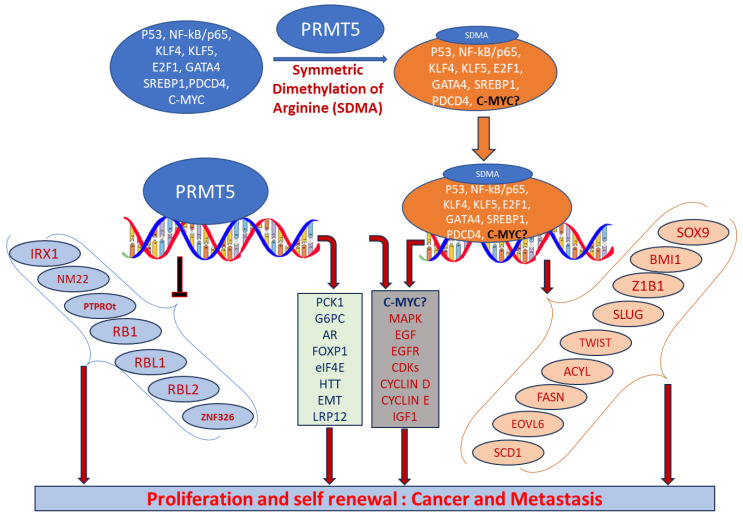
Biological functions of PRMT5 that regulate cellular processes. Elevated expression of PRMT5 can cause post-translational modification of several transcription factors by symmetrically dimethylating arginine residues of proteins and regulate the expression of their corresponding targeted genes. When recruited to the promoter regions of precise target genes in the nucleus, they can promote cell proliferation and tumorigenesis.

**Figure 3 cancers-15-05855-f003:**
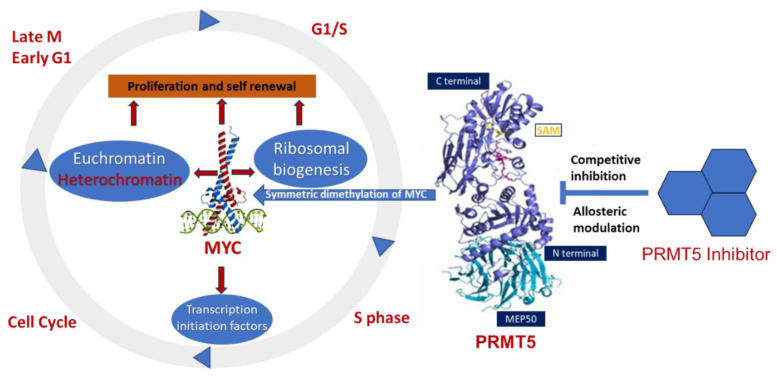
Overexpression of PRMT5 causes symmetric demethylation and stabilization of MYC, leading to reduced apoptosis and enhanced cell proliferation. As indicated, various steps in this process can be modulated by PRMT5 inhibitors.

**Table 2 cancers-15-05855-t002:** Pharmacologically active PRMT5 inhibitors.

Compound Name	Structure	Function	IC50 In Vitro	In Vivo Activity	References
JNJ64619178	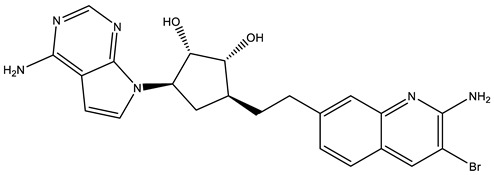	Dual SAM/substrate competitive	0.2 nM	Antitumor effect in lung cancer, AML, non-Hodgkin lymphoma cell line mouse xenograft	[108]
PF06939999	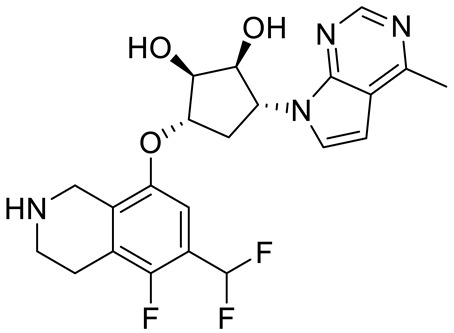	SAM competitive	3.3 nM	Antitumor effect in lung cancer	[109]
GSK3235025 EPZ015666	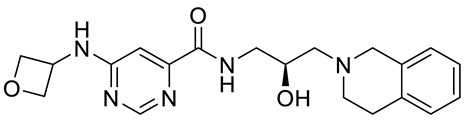	Substrate competitive	22 nM	Antitumor effect in MCL, MM, AML, GBM, and bladder cell line mouse xenografts and in a TNBC PDX mouse model	[110,111,112,113,114]
GSK591 (EPZ015866)	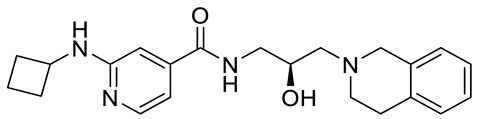	Substrate competitive	4 nM	Antitumor effect in glioblastoma	[110]
GSK3326595	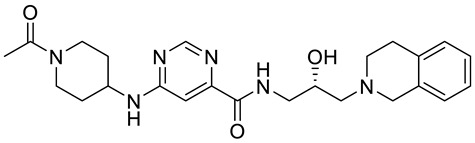	Substrate competitive	6.2 nM	Antitumor effect in non-Hodgkin lymphoma cell line mouse xenograft and antitumor effect in a DLBCL PDX mouse model	[115,116,117,118]
AMG 193	Structure undisclosed	MTA cooperative inhibitor	NA	Antitumor effect on advanced/metastatic solid tumors	[119]
PRT543	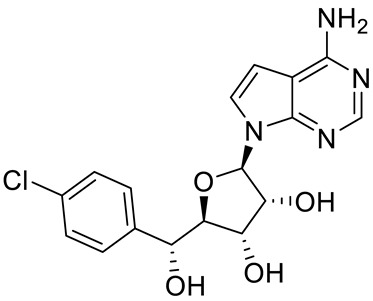	SAM competitive	10.8 nM	Antitumor effect on advanced solid tumors and hematologic malignancies	[120]
PRT382	Structure undisclosed	SAM competitive	2.8 nM	Antitumor effect on hematological tumors	[85]
PRT811	Structure undisclosed	SAM competitive	3.9 nM	Antitumor effect on advanced solid tumor, Glioblastoma, CNS Lymphoma	[121]
TNG908	Structure undisclosed	MTA cooperative inhibitor	110 nM	Antitumor effect on Glioblastoma,	[122]
MRTX1719	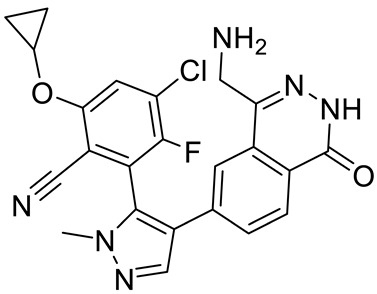	PRMT5–MTA complex inhibitor, MTA competitive	12 nM	Antitumor effect on solid tumor	[123,124]
LLY-283 (C220)	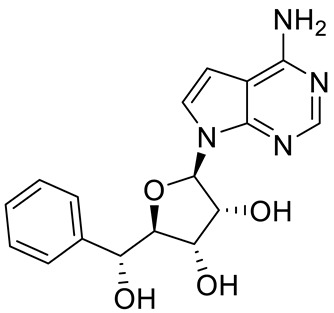	SAM competitive	22 nM	Reduced acute graft versus host disease incidence in mice, antitumor effect in MPN xenografts	[74,125,126]
Compound1a	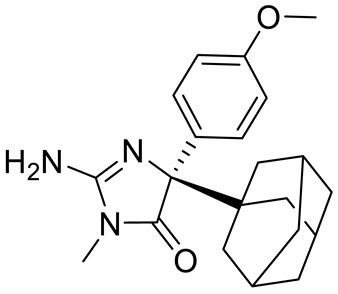	Allosteric modulator	16 nM	Antitumor effect in breast cancer	[127]
CMP5	Structure undisclosed	SAM competitive	25 µM	Antitumor effect in breast cancer and glioblastoma	[72]
JBI-778	Structure undisclosed	Substrate competitive	27 to 700 nM	Antitumor effect in glioblastoma	[128]
SH3765	Structure undisclosed	Substrate competitive	NA	Antitumor effect on advanced malignant tumors, including solid tumors and non-Hodgkin lymphoma	[129]
SCR6920	Structure undisclosed	Substrate competitive	NA	Antitumor effect on advanced malignant tumor including solid tumor and non-Hodgkin lymphoma	[129]

Abbreviations: AML, acute myeloid leukemia; MCL, mantle cell lymphoma; MM, myelomonocytic leukemia; GBM, glioblastoma; TNBC, triple-negative breast cancer, PDX, patient-derived xenograft; DLBCL, diffuse large B cell lymphoma; CNS, central nervous system; MPN, myeloproliferative neoplasm; nM, nano molar; NA, not available; SAM, S-adenosylmethionine; MTA, methylthioadenosine.

**Table 3 cancers-15-05855-t003:** PRMT5 inhibitors in clinical trials.

ClinicalTrials.gov Identifier	Name of Inhibitor	Status	Disease
NCT03573310	JNJ64619178	Phase I	Neoplasm solid tumors, non-Hodgkin lymphoma, and myelodysplastic syndrome
NCT03854227	PF06939999	Phase I	Advance and metastatic solid tumors
NCT03614728	GSK3326595	Phase I and II	Metastatic solid tumors and acute myeloid leukemia
NCT02783300	GSK3326595	Phase I	Solid tumors and non-Hodgkin lymphoma
NCT04676516	GSK3326595	Phase II	Early-stage breast cancer
NCT03886831	PRT543	Phase I	Advanced solid tumors and hematological malignancies
NCT05275478	TNG908	Phase I and II (recruiting)	Locally advanced solid tumors
NCT04089449	PRT811	Phase I (recruiting)	Advanced solid tumors, recurrent glioma, and CNS lymphoma
NCT05245500	MRTX1719	Phase I and II (recruiting)	Mesothelioma, NSCLC, malignant peripheral nerve sheath tumors, solid tumors, and pancreatic adenocarcinoma
NCT05094336	AMG 193	Phase I and II (recruiting)	Advanced MTAP-null solid tumors
NCT05528055	SCR6920	Phase I (recruiting)	Advanced malignant tumors
NCT05015309	SH3765	Phase I (not yet Recruiting)	Advanced malignant tumors

## Data Availability

This article does not report any original data.

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
