# Peer review of "PRMT5 as a Potential Therapeutic Target in MYC-Amplified Medulloblastoma"

_cancers, 2023, doi:10.3390/cancers15245855_

Round 1
Reviewer 1 Report
Comments and Suggestions for Authors
MYC amplification/or overexpression is most common in Group 3 medulloblastoma and positively associated with poor clinical outcomes. Recently, protein arginine methyltransferase 5 (PRMT5) overexpression has been shown to be associated with tumorigenic MYC function in cancers, particularly in brain cancers such as glioblastoma and medulloblastoma. PRMT5 regulates oncogenes including MYC, that are often deregulated in medulloblastoma. However, the role of PRMT5-mediated posttranslational modification in stabilization of oncoproteins remains poorly understood. The potential impact of PRMT5 inhibition on MYC makes it an attractive target in various cancers. PRMT5 inhibitors are the promising class of anticancer drugs demonstrating preclinical and preliminary clinical efficacies. Here, we review the publicly available preclinical and clinical studies on targeting PRMT5 using small molecule inhibitors and discuss the prospects of using them in medulloblastoma therapy. But I have several following comments:
1. Abbreviations should be defined when they first appear in the text. Such as "GLI1", "AML"...
2. Please add the line number to the article, so that the reviewer can find the line number and locate it, so that the author can revise it later.
3. The structures of the PRMT5 inhibitors reported here should also be provided to facilitate the development of more effective drugs by pharmacologists accordingly..
4. If Figures 1 and 2 are a copy of the reported picture or made with software, please apply for copyright or indicate the quotation.
5. Tables should use a standard three-line table and should not span pages.
6. Some numbers and quantifiers should have Spaces between them, please check the full text and modify according to the rules. Such as "0.2nM" in Table 2, ....
7. "25uM" should be " 25 μM", please double check the whole text and correct them.
8. In Table 2 "in vitro" and "in vivo" should be be in italics
9. The nucleic acid sequences (including gene names, regulatory sequences, and primer names) should be in italics.
10. Please unify the format of references in the article, including the author's name, the case of words in the title of the article, the writing of the name of the journal, and the page number.
Comments on the Quality of English LanguageMinor editing of English language required.
Reviewer 2 Report
Comments and Suggestions for Authors
The article presents the role of PRMT5 in cancer, in particular in medulloblastoma. The article is properly organized and gives a proper view of the topic.
Suggestion: The authors should indicate the possibilities of combined treatment of PRMT5 with other drugs (e.g. chemotherapy, anti-PDL-1 drugs, or anti-EGFR). Also indicate studies in which such attempts were made and comment on the possibility of concomitant use of PRMT5 and PRMT1 inhibitors
Reviewer 3 Report
Comments and Suggestions for Authors
The authors submit a review on PRMT5 as a potential therapeutic target in MYC-amplified medulloblastoma. They conclusively summarise that MYC amplification or overexpression is common in Group 3 medulloblastoma and associated with poor clinical outcomes. In context with MYC, protein arginine methyltransferase 5 (PRMT5) overexpression has been shown to be associated with tumorigenic MYC function in medulloblastoma, and it has been demonstrated that PRMT5 regulates oncogenes including MYC. However, the role of PRMT5-mediated posttranslational stabilization of oncoproteins remains poorly understood. PIn addition, RMT5 inhibitors are in development demonstrating preclinical and preliminary clinical efficacies.
Major points:
The manuscript should be restructured within its sections. E.g., in the introduction, a possible way would be to start with expression of PMRT5, continue with cellular and sub cellular localisation, then coming to its functions in development, normal tissue and cancer, and ending with mechanisms od function and data in animal models and humans. As it is now, the subsections of the manuscript jump back and forth.
I would suggest to reorganize Table 1 according to function in the first order. In addition, formatting can be improved.
Minor points:
Page 1, line 36. SHH is subdivided into p53 mutant and p53 wild type.
Introduction: it should be mentioned that MYC alterations predominantly occur in children, and almost never in adults.
I would suggest to delete the company names in Table 3.
Comments on the Quality of English LanguageThe english language must be improved. There are often lengthy and redundant sentences, missing words a.s.o.. In addition, in some parts, the reader can only guess what is meant by te authors, as sentences are incomplete or so long that it becomes unclear what content is connected. Review by a native speaker is therefore mandatory.
Reviewer 4 Report
Comments and Suggestions for Authors Kumar et al present an overview of PRMT5 targeting in MYC-driven Group 3 MB and summarize the currently available clinical trials, pharmacologically active PRMT5 Inhibitors, and future directions & challenges as a therapeutic. Introduction- Could touch on Group 4 as they did WNT, SHH, and Group 3 in the first paragraph
- Figure 1 could be made simpler/more intuitive alternatively figure legend could be lengthened and made more descriptive (does the left hand components directly increase expression of PRMT5 to enhance SDMA, does the left hand components drive translocation of PRMT5 to the nucleus? etc)
- a figure could be included to accompany structural descriptions (line 86-100)
- Figure 2 not super accessible colouring, maybe make the PRMT5 and MYC colocalization complex more clear (could label it), not sure the significance of the cell cycle checkpoints are in this specific figure
- For all Figures, resolution and quality could be improved
- Great points raised about challenges to PRMT5 therapies including BBB permeability, PKPD, toxicity (essentiality in NSCs), lack of mechanistic insight perhaps another subheading "limitations to PRMT5 therapy" could be made so that the discussion isn't so long/broad
Comments on the Quality of English Language
Overall spelling and grammar should be extensively checked throughout the manuscript (ie line 145 why is melanoma capitalized?, line 147 cellular localization- did they mean cytoplasmic?, line 148 did they mean confirmed not conformed?, line 152 play crucial role, line 157 grammar, etc)
Round 2
Reviewer 1 Report
Comments and Suggestions for Authors
The authors have addressed all my concerns. I recommend accepting it in current form.
Reviewer 3 Report
Comments and Suggestions for Authors
Authors have revised the manuscript according to the reviewers suggestions.
Comments on the Quality of English LanguageEnglish language is fine now.